# Māori Health, Wellbeing, and Disability in Aotearoa New Zealand: A National Survey

**DOI:** 10.3390/ijerph22060829

**Published:** 2025-05-23

**Authors:** Tristram R. Ingham, Bernadette Huatau Jones, Meredith A. Perry, Andrew Sporle, Tom Elliott, Paula Toko King, Gabrielle Baker, Barry Milne, Tori Diamond, Linda Waimarie Nikora

**Affiliations:** 1Department of Medicine, University of Otago Wellington, Wellington 6021, New Zealand; bernadette.jones@otago.ac.nz; 2Foundation for Equity and Research New Zealand, Wellington 6012, New Zealand; 3School of Physiotherapy, University of Otago, Dunedin 9016, New Zealand; meredith.perry@otago.ac.nz; 4Department of Statistics, University of Auckland, Auckland 1010, New Zealand; a.sporle@auckland.ac.nz (A.S.); tom.elliott@auckland.ac.nz (T.E.); 5iNZight Analytics Ltd., Auckland 1010, New Zealandtori@inzight.co.nz (T.D.); 6Te Rōpū Rangahau Hauora a Eru Pōmare, Department of Public Health, University of Otago Wellington, Wellington 6021, New Zealand; paula.king@otago.ac.nz; 7Baker Consulting Ltd., Wellington 6011, New Zealand; gabrielle@bakerconsulting.co.nz; 8COMPASS Research Centre, University of Auckland, Auckland 1010, New Zealand; b.milne@auckland.ac.nz; 9Ngā Pae o te Māramatanga, University of Auckland, Auckland 1010, New Zealand; l.nikora@auckland.ac.nz

**Keywords:** indigenous health, Māori wellbeing, health disparities, Māori worldview, health inequities, cultural identity, disability and health, intersectionality in health

## Abstract

Māori, the Indigenous people of Aotearoa New Zealand, experience wide-ranging inequities compared with non-Māori. This survey aimed to explore the holistic health, wellbeing, and disability experiences of New Zealand’s Indigenous Māori population from a Māori worldview, addressing gaps in culturally relevant data often overlooked by standard health surveys. A robust cross-sectional survey was conducted with 7359 participants of Māori descent using Kaupapa Māori Research principles. Data were analysed using the Te Pae Māhutonga framework, a Māori health promotion model. Participants demonstrated strong cultural identity, with 32.3% understanding spoken Māori fairly well and 97.3% defining a broad non-nuclear concept of whānau (family). While over half reported high life satisfaction, 58.4% experienced discrimination, mainly based on ethnicity and appearance. Access to healthcare revealed that 32.6% were unable to contact a general practitioner due to cost. Socioeconomic challenges were prevalent; nearly a quarter borrowed from family or friends to meet daily living costs, and over a third economized on fresh produce to save money. This study reveals significant gaps in mainstream health data and demonstrates that a culturally aligned, methodological approach is feasible and crucial for informing policies that address the needs and rights of Māori, as guaranteed under Te Tiriti o Waitangi. These results could inform global, indigenous research addressing culturally relevant health, wellbeing and disability inequities.

## 1. Introduction

Historical and contemporary data have persistently demonstrated that Māori suffer a disproportionate burden of inequities in health, wellbeing, and disability compared with non-Māori in Aotearoa New Zealand (NZ). These inequities represent a loss of human capital and impose significant personal, community, and social costs on Māori and the whole country. Addressing inequities, from a purely health economic perspective, has the potential to save over NZD 800 million per year [1,2,3]. These current losses could be more productively invested in health and social services. From both human rights and legal perspectives, the government has an obligation to monitor inequities and proactively eliminate them through policy and system change. These obligations are non-negotiable under Te Tiriti o Waitangi, NZ’s foundational document. These requirements are reinforced by NZ’s ratification of international human rights conventions, particularly the Universal Declaration on Human Rights and the United Nations Declaration on the Rights of Indigenous People (UNDRIP) [4], and the United Nations Convention on the Rights of Persons with Disabilities (UNCRPD) [5].

As the Indigenous peoples of NZ, Māori possess holistic models of health, wellbeing, and disability that diverge from the prevailing Western biomedical frameworks [6,7]. In contrast, these Māori models are fundamentally collective in nature, emphasizing the interconnectedness of individuals within their social and environmental contexts. These models encompass vital cultural practices and concepts, including tikanga (protocols), te reo Māori (the Māori language), mātauranga (knowledge), whakapapa (genealogy), and social connectedness [7]. They explicitly recognize the inseparable link between individual health and wellbeing and broader holistic elements, such as whānau (extended family), environment, and spirituality, which are not fully captured in official statistical surveys [7].

To direct policy, it is important that valid quality data be collected; however, to date, data on the constructs of health, wellbeing and disability are typically constructs defined by governments and policymakers. A prime example of this is in the area of disability data, where the Waitangi Tribunal (a permanent Commission of Inquiry) found in 2021 that the government had a “practical absence of quality data on tāngata whaikaha” (Māori with lived experience of disability) [8] (p. 112), which negatively impacted the development and roll-out of a national COVID19 vaccination strategy. This is the result of decades of Māori health, wellbeing, and disability data being collected and presented using a biomedical perspective and excluding any meaningful sociocultural and economic wellbeing understanding. Te Kupenga is the only post-censual government survey specifically of Māori, providing social, cultural, and economic wellbeing perspectives [9]; however, this survey includes only one overall health question and uses linked Census data for disability information.

Government-administered surveys typically use data collection processes, analytical lenses, and interpretation of results based on the dominant colonial perspective. This approach does not align with a holistic Māori worldview; it largely reflects government priorities with the exclusion of Māori aspirations; and the reliance on administrative data means that the data collection and analysis are often removed from a specific context. Consequently, the results are not reliable interpretations of Māori health, wellbeing, or disability.

There are major demographic differences in the Māori population. For example, the age structure is markedly different from that of the NZ European population, with a median age of 25.4 years for Māori compared with 41.4 for NZ Europeans [10]. To understand these important differences, large surveys of Māori using a Te Ao Māori approach are required to provide a more nuanced understanding of Māori health, wellbeing, and disability. Currently, due to the sample size of Māori in national datasets, Māori are often considered a homogeneous group, with their data used particularly for comparisons with total population data [11]. The annual NZ Health Survey (NZHS) provides self-reported health, access utilization of healthcare, disability-related functional impairment, and sociodemographic data, but excludes collective wellbeing and cultural aspects. The 2022/23 NZHS collected data from 6799 adults and 2686 children, of which Māori comprised 1292 adults and 657 children [12]. This sample size is not large enough to describe diversity in health status measures within the Māori population.

Relationships between health, wellbeing, and disability are complex, and the current methods of collecting, analysing, and interpreting data miss culturally relevant and nuanced sub-group outcomes [11]. These surveys contribute to the portrayal of Māori from a deficit paradigm, i.e., only as experiencing health inequities in comparison with other demographic groups, rather than an independently rich and diverse population in their own right. Current Māori-specific health and disability policies are unlikely to be fit for purpose. Novel ways of exploring Māori health, wellbeing, and disability data, including sociocultural perspectives, will contribute to a more contextually relevant understanding of the determinants of health inequities.

We are a Māori-led collective of researchers whose research is situated within a Kaupapa Māori paradigm that critiques those underlying power dynamics of colonization and coloniality that drive and maintain Māori health and other inequities [13]. This article shares the findings of a novel cross-sectional survey of adults of Māori descent from across NZ and aims to demonstrate the value of a survey designed to describe the wellbeing of NZ’s Indigenous population from a Māori worldview.

## 2. Methods

### 2.1. Study Design

This study comprised a nationally representative, cross-sectional survey of adults of Māori descent across NZ. The survey design has been published previously (available here), but core features are summarized below [14]. Eligible participants were NZ citizens or residents aged 18 years and older, registered on either the General or Māori electoral rolls in 2021, and self-identifying as being of Māori descent (N = 527,598). A random sample of 70,155 participants was drawn, excluding those who had an overseas mailing address. The survey was conducted primarily online using Qualtrics Experience Management software (version designed in August 2022), and in accordance with the Checklist for Reporting Results of Internet E-Surveys (CHERRIES). To ensure accessibility, telephone interviewer-administered and self-completed paper options were also available in English or te reo Māori. Recruitment was conducted in two tranches, from July 2023 to December 2023, with participants in each tranche receiving a personalized invitation letter and two reminder postcards.

### 2.2. Study Size

Based on previous national survey response rates, a sample size of at least 70,000 invitations was calculated to achieve approximately 8000 responses, given an anticipated 14% response rate and an 80% survey completion rate. This sample size was estimated to ensure the study would be adequately powered to identify key health-related outcomes among Māori. The sampling error was calculated per a simple random sample with post-stratification weights applied. The actual margin of error was 1.1% overall.

### 2.3. Bias

We included robust processes to ensure the sample was generalizable to the Māori descent population. This study was grounded in Kaupapa Māori Research (KMR) methodology, culturally appropriate, and accessible in terms of language and formats. Furthermore, robust validation measures were deployed and two-stage weighting was adopted to adjust to the electoral roll sample for non-response bias, and then to the Administrative Population Census (APC) [15], for under-coverage in the sampling frame [14].

### 2.4. Holistic Framework of Māori Wellbeing—Te Pae Māhutonga

We present our study outcomes under the framework of a Māori model of health promotion to approximate the holistic nature of Te Ao Māori (Māori worldview) conceptualizations of health and wellbeing. Te Pae Māhutonga (the Southern Cross star constellation), a culturally and historically significant Māori model of health developed by Sir Mason Durie [16], guides health initiatives by reflecting six elements applicable to Māori and broader NZ:Mauriora: Access to Te Ao Māori

Mauriora, the flourishing of Māori identity, is essential for wellbeing. Embracing and strengthening Māori identity promotes positive health outcomes, highlighting the importance of cultural revitalization for overall wellbeing [16]. In this domain, we include various aspects of cultural identity such as cultural knowledge, practices, and language.

“*Mauriora rests on a secure cultural identity. Good health depends on many factors, but among indigenous peoples the world over, cultural identity is considered to be a critical prerequisite*”.(Durie 1999) [16] p. 2

2.Waiora: Environmental Resources

Waiora, the flourishing interconnectedness between people and their environment, is essential for wellbeing. It fosters a deep connection to the natural world, encompassing the cosmic, terrestrial, and aquatic realms. In this domain, we include connections to wellbeing resources such as physical resources (e.g., environment), social resources (e.g., culture, whakapapa, and whānau), and system (e.g., community and marae).

“*Waiora is linked … to the external world and to a spiritual element that connects human wellness with cosmic, terrestrial and water environments*”.(Durie 1999) [16] p. 3

3.Toiora: Healthy Lifestyles

Toiora, the active pursuit of healthy lifestyles, empowers individuals to reach their full potential. By making informed choices about nutrition, exercise, and safety, individuals can mitigate risks and prevent health issues. Embracing Toiora not only enhances personal wellbeing, but also contributes to the collective prosperity of the Māori community and the nation. In this domain, we include general measures of health and wellbeing, life satisfaction, and self-reported disability.

“*Major threats to health come from the risks that threaten health and safety and have the capacity to distort human experience*”.(Durie 1999) [16] p. 3

4.Te Oranga: Participation in Society

Te Oranga, active participation and equitable access within society, is a cornerstone of wellbeing. It empowers individuals through fair access to essential services, education, employment, and decision-making processes. When Māori actively participate and contribute their voices, the entire society benefits from increased diversity, resilience, and innovation. In this domain, we include items exploring societal influence on wellbeing, particularly healthcare access, societal inclusion, and discrimination.

“*Wellbeing is … the goods and services which people can count on, and the voice they have in deciding the way in which those goods and services are made available. Te Oranga is dependent on the terms under which people participate in society and on the confidence with which they can access good health services, or the school of their choice*”.(Durie 1999) [16] p. 4

5.Ngā Manukura: Leadership

Ngā Manukura, effective leadership, is a collaborative process that draws upon diverse skills and perspectives. It recognizes and empowers local leaders within communities, fostering a relational approach and building alliances between different groups. By uniting diverse expertise and forging connections, Ngā Manukura can effectively drive positive change and achieve shared goals. In this domain, we include items exploring both social and professional leadership.

“*Leadership … should reflect a combination of skills and a range of influences. Regardless of technical or professional qualifications, unless there is local leadership it is unlikely that a health promotional effort will take shape or bear fruit*”.(Durie 1999) [16] p. 5

6.Te Mana Whakahaere: Autonomy

Te Mana Whakahaere, the exercising of autonomy and self-determination, empowers communities to shape their own wellbeing. It fosters a sense of ownership and control over decision-making processes, enabling communities to prioritize their unique aspirations, values, and initiatives that lead to more effective and sustainable outcomes. In this domain, we include factors such as economic freedom and material wellbeing that act as precursors to the fulfilment of autonomy and self-determination.

“*Communities … must ultimately be able to demonstrate a level of autonomy and self-determination in promoting their own health*”.(Durie 1999) [16] p. 6

### 2.5. Variables

#### Participant Demographics

Gender was collected using the question from the 2021 What about me? nationwide survey of youth across New Zealand [17] and consistent with the StatsNZ statistical standard for reporting gender [18]. Age group was collected as the year of birth and reported using 15-year bands. The lowest age group was 18 years, while the highest age group aggregated participants 75 years and above. Ethnicity was collected in accordance with NZ Ethnicity Data Collection Standards v2.0. [19], reporting Level 1 total response ethnicity. Iwi is reported as the total response and categorized consistent with the StatsNZ iwi data standards used in both the Census and Te Kupenga [20]. Iwi was grouped into 18 categories with Iwi affiliation, while geographically bounded, not necessarily related to the geographic region of residence.

Region is reported as 1 of 16 geographic regions defined by StatsNZ per Census 2018. Urbanicity was derived from the address recorded on the electoral roll. The StatsNZ Urban Rural classification V1.0.0 was used to allocate each respondent to a rural or urban category based on the mesh block corresponding to the address [21]. All values were collapsed to three categories: major urban, minor urban, and rural.

*1.* 
*Mauriora: Access to Te Ao Māori*


Māori language use and fluency were captured using Likert response questions from the StatsNZ Te Kupenga Survey [9]. These questions included an ability to speak in day-to-day conversation and understand spoken reo, and ability to read and write in te reo Māori. Cultural participation included a variety of traditional and contemporary opportunities to learn, engage with, express, and participate in Te Ao Māori contexts (e.g., cultural events, traditional oratory and song, cultural or marae-based gatherings, and traditional healing practices). Questions were selected from Te Kupenga Survey [9] and reported as dichotomous outcomes over a 12-month recall period.

*2.* 
*Waiora: Environmental Resources*


Connection to Whakapapa (Lineage) and Whenua (Land)

Māori culture is widely acknowledged as a collectivist, tribal culture whereby kinship relationships and connection to the environment are seen as integral components of a person’s wellbeing [7]. Connection to whakapapa or genealogical ancestry was collected as *knowledge* of iwi (tribe) and hapū (sub-tribe), tribally linked prominent geographical features (mountains and rivers), tīpuna (ancestors), waka (canoe migrating from Hawaiiki), and *feeling* of connection to traditional lands referred to as tūrangawaewae (place to stand). We limited the scope to cultural knowledge and feeling of connection, thus circumventing measures of physical connection to avoid mixing those things with the ability to connect, which is likely confounded by socioeconomic and health-related limitations. Questions were drawn from Te Kupenga Survey for comparison.

Marae access was collected over two recall periods (ever; in the last 12 months) for both urban and ancestral marae. Whānau, or extended family groupings, are the fundamental unit of Māori society. As a result of colonialism and urbanization, there are increasingly diverse interpretations of whānau, with many non-Māori applying the Western concept of nuclear family [22]. Whānau connection was collected according to self-definition and self-assessed ratings of wellness, connection, and amount of contact using Likert responses. Dichotomous outcomes in relation to traditional food gathering, weaving, carving, and environmental protection activities were assessed over a 12-month recall period using questions from Te Kupenga Survey [9].

*3.* 
*Toiora: Healthy Lifestyles*


Life satisfaction was collected from all participants using a well-validated 5-point Likert response (‘very dissatisfied’ to ‘very satisfied’). Quality of life was collected using the extensively validated SF-12 questionnaire. To minimize respondent burden, the physical functioning subdimension questions were not asked directly but derived from the equivalent Washington Group Extended Set (WGES) questions. The SF-12 ‘moderate activities’ referred to health limitations with ‘activities you might do during a typical day … such as moving a table, a vacuum cleaner, or bowling’. WGES-equivalent responses were assigned based on the highest limitation from ‘upper body‘ strength or ‘self-care’ questions. Similarly, the SF-12 ‘climbing stairs’ question referred to health limitations with ‘climbing several flights of stairs’. WGES equivalent responses were derived based on the limitation ‘walking or climbing steps’.

Self-reported disability was collected using the standard question from NZHS 2023 [12], including all health conditions, and long-term and mental health conditions, which affect a person’s ability to carry out everyday activities. Activity limitation was collected using the Washington Group Extended Set of Questions on Functioning (WGEF), a subset of which, the Washington Group Short Set (WG-SS), is the internationally accepted benchmark for comparisons and longitudinal monitoring of a sub-population of people with severe impairments likely at risk of experiencing ableism/disablism in unaccommodating environments [23].

*4.* 
*Te Oranga: Participation in Society*


Data were collected for healthcare access as a key determinant of health. Questions were selected from NZHS questions regarding GP enrolment, healthcare contact in the last 12 months, unmet health need, unmet dental need, unmet mental health need, and cost barriers. All responses were dichotomous, except for dental access, which was collected semi-quantitatively.

A bespoke set of questions on social inclusion was developed in partnership with our community steering groups through hui (meeting), wānanga (workshop), and whakawhiti kōrero (negotiating) sessions. Each question used a 3-point Likert response (‘none’; ‘a little’; ‘a lot’) to self-reported sense of inclusion in three spheres of engagement: your community, Te Ao Māori, and broader NZ. We also collected the extent to which NZ policies, services, supports, and attitudes impact the person’s ability to perform day-to-day activities and achieve life goals.

Discrimination data were collected from all participants using ‘ever discrimination’, [3] from Te Kupenga Survey. ‘Ever discrimination’ was favoured over recent discrimination as a better proxy for long-term health and wellbeing, as it is likely to have impacted reported outcomes such as educational, employment, other health, and social opportunities.

*5.* 
*Ngā Manukura: Leadership*


Data collected in this domain included current employment (dichotomous response) and occupation as classified by the Australian and New Zealand Standard Classification of Occupations (ANZSCO) v1.30 [24].

*6.* 
*Te Mana Whakahaere: Autonomy*


NZDep2018 is a metric that measures the level of socioeconomic deprivation for people in each small census area of about 50 households, based on nine Census variables [25]. Quintile 1 represents areas with the *least* deprived scores; Quintile 5 represents areas with the *most* deprived scores. NZDep2018 was calculated based on the mesh block of the address provided in the electoral roll, which served as our recruitment mailing address [14]. Individual income was collected using the Census 2018 income question. Data are reported in NZD 5000 bands up to NZD 40,000, NZD 10,000 bands up to NZD 70,000, then a NZD 30,000 band to NZD 100,000, and a NZD 50,000 band to NZD 150,000, with all higher income collated into NZD 150,000 or more. These variable width bands reflect the skewed distribution of income and income-derived benefit thresholds.

Questions were selected from the Material Wellbeing Questionnaire (MWQ) of the Household Economic Survey [26]. The MWQ asks about ownership of items, performing certain activities, the extent that people economize, self-rating life satisfaction, and whether income meets every day needs. We report dichotomous wellbeing outcomes exploring income adequacy to meet basic needs: ownership and participation (e.g., owning: suitable shoes; affording: a meal out); economizing (e.g., going without fruit and vegetables, postponing healthcare); financial freedom/restriction (e.g., ability to pay an unavoidable expense); and financial strain (e.g., inability to pay bills, rent, or receiving welfare).

### 2.6. Statistical Method

Our sample-weighting protocols and related regression model are available in our published methodology [14]. Weighting was based on gender, age group, region, occupation, NZDep2018 quintile, and urbanicity. Data are presented as weighted percentages of the survey population, along with estimates of total APC-Māori descent population (*n* = 526,476), each with 95% confidence intervals.

## 3. Results

Responses were received from 7359 of the 66,175 eligible participants (11.1% response rate; 92.7% via online completion, 5.4% via telephone, and 2.0% self-completed on paper). Overall, 7230 participants (99.6%) provided sufficient ID and demographic details to be verified as eligible unique individuals and were therefore included in the analysis. Of those 7230, 6774 answered the survey fully, giving a survey completion rate of 93.7% [27]. A flowchart of recruitment and eligibility has been published previously [14].

Table 1 shows the gender, age group, and ethnicity variables; 51% of participants identified as female and 56.0% were aged 18 to 44 years. All participants were of Māori descent. Of these, 95.4% identified as Māori, while 56.4% also identified as European, and 7.0% as Pacific peoples. Table 1 also presents the geographic regions where participants lived. These data reflect the distribution of the Māori population, weighted to the APC. Most participants (60.2%) live in major urban environments; 16.3% live in rural areas.

Table 2 shows a range of iwi (tribal) representation with 26.7% of participants from Te Tai Tokerau/Tāmaki-Makaurau (Northland/Auckland) Region, 11.4% from Te Wai Pounamu (South Island) and 0.4% from Rēkohu/Wharekauri (Chatham Islands) imi/iwi.

*1.* 
*Mauriora: Access to Te Ao Māori*


Table 3 shows that 16.9% of participants report being able to speak te reo Māori at least fairly well; 33.2% can understand spoken te reo Māori at least fairly well, 29.6% can read te reo Māori at least fairly well, and 20.8% can write it at least fairly well. Almost half (45.8%) of the participants reported finding it easy or very easy to access Māori language learning resources.

Table 4 reports cultural participation amongst Māori; 61.9% have sung a waiata (Māori song), performed a haka, given a mihi (Māori introduction) or speech, and 57.8% have said a karakia (prayer). There was lower collective participation in activities; for example, 15.1% had taken part in a kapa haka (cultural performance).

*2.* 
*Waiora: Environmental Resources*


Table 5 presents data on connections to environmental and cultural resources. The table shows that most participants (89.2%) reported knowing their iwi; 70.0% reported knowing their ancestral marae; and 30.5% of the participants felt strongly or very strongly connected to their tūrangawaewae. A proportion of 96% of participants report having been to a marae, with 48.0% having been to a marae in the last 12 months; 59.3% have been to any of their ancestral marae; and 29.8% have been to their ancestral marae in the last 12 months. Table 5 also presents results on traditional resource-gathering practices; 44.7% of participants reported having gathered traditional Māori kai (food) or materials for carving/weaving in the last 12 months.

Table 6 reports meaningful social connections, including whānau identity and connectedness and level of recent contact with whānau. A high proportion of participants considered a broad concept of family as their whānau; over 80% defined aunts/uncles, cousins, nephews/nieces, and other in-laws as whānau; and 45.9% defined friends as whānau. Most participants (88.7%) reported that their whānau were doing somewhat or very well, and 87.7% felt somewhat or very connected to their whānau. Over half (52.8%) had seen whānau at least weekly over the previous 4 weeks; 44.2% reported that their level of contact was insufficient.

*3.* 
*Toiora: Healthy Lifestyles*


Table 7 reports on general life satisfaction and general health. Half of the participants (49.9%) report feeling satisfied or very satisfied with their life right now; 71.4% report having at least good general health. Table 8 also presents data on self-reported disability and activity limitations. Over one-fifth (21.2%) of participants self-report having a disability, long-term condition, or mental health condition that limits their ability to carry out everyday activities. Activity limitation (‘a lot of difficulty’ or ‘cannot do at all’) in at least one functional domain was reported by 15.1% of participants.

Table 8 reports data on health-related quality of life in terms of both physical and mental health. A large proportion of participants reported accomplishing less than they would like (some, most, or all of the time) as a result of their physical health (42.9%), and their emotional health (35.3%) respectively. Concerningly, 31.5% of participants felt downhearted or depressed some, most, or all of the time, equating to 166,340 adults of Māori descent nationwide.

*4.* 
*Te Oranga: Participation in Society*


Table 9 shows access to health system resources. Most participants (88.0%) report having access to a GP and (74.0%) attending that clinic in the last 12 months. Conversely, 34.6% reported not being able to access their usual GP within 24 h of needing care, and 32.6% reported not contacting a GP due to cost. Almost a quarter (24.0%) of participants had not received dental care in the last 5 years; 59.1% avoided dental care due to cost. In the past 12 months, 22.0% of participants reported unmet need for professional service for psychological or mental health needs.

Table 10 shows that 20.6% of participants feel well included within their community, 13.0% in Te Ao Māori, and 16.5% in broader NZ society. NZ societal structures, policies and attitudes were reported to have a lot of impact on 29.6% of the participants’ abilities to do day-to-day activities and 33.6% of the participants’ abilities to achieve their life goals.

Table 11 shows that over half of the participants (58.4%) report having been discriminated against during their life. Most (74.8%) of those reporting discrimination listed race or ethnic group as a basis for discrimination, with nearly half of discriminated participants reporting either skin colour or appearance as reasons for discrimination. Discrimination was reported in a diverse range of contexts, but the highest proportion (68.7%) was at school.

*5.* 
*Ngā Manukura: Leadership*


Table 12 reports aspects of leadership in terms of community and professional leadership opportunities. Notably, 11.2% of all participants reported taking part in environmental planning on behalf of their iwi, hapū, or marae. Just over half of participants (52.0%) reported being in current employment, with the highest occupational category being professionals (10.5%); 11.5% were students; and 13.0% were not in the labour force.

*6.* 
*Te Mana Whakahaere: Autonomy*


Table 13 demonstrates that participants were overrepresented among high deprivation NZDep2018 census areas, with 38.7% in the highest deprivation quintile; 16.4% of participants reported an individual annual income NZD 15,000 or less, and 33.6% of participants earned between NZD 15,000 and NZD 50,000.

Material wellbeing is reported in Table 14 (income adequacy) and Table 15 (economizing, financial freedoms, and financial capacity).

Participants reported (Table 14) generally high rates of vehicle access (83.0%) and internet connectivity (83.6%). Just over half (53.3%) of participants have home contents insurance, with fewer than half (47.2%) having a domestic holiday annually and fewer than a quarter (24.2%) having an international holiday triennially.

Many participants report economizing activities in the last year (Table 15). Over a third (36.3%) of participants have put up with feeling cold or going without fruit and vegetables (34.8%) to keep costs down; 56.6% have deferred dental care; and almost half of participants have deferred household maintenance activities due to cost. Nearly two-thirds of participants (63.8%) report having adequate financial reserves to be able to incur an unexpected cost of NZD 500; a quarter (24.4%) need to seek financial assistance or loans from family and friends to meet every day needs; and almost a fifth of participants (18.3%) report receiving charity assistance or welfare in the last 12 months.

## 4. Discussion

This survey is the largest ever community-initiated cross-sectional survey specifically of Māori health, wellbeing, and disability, and the first non-governmental study in two decades to rigorously examine the intersectionality of a broad, holistic range of domains. It offers unprecedented insights into indigenous health dynamics that are essential for effective public health policy. Importantly, this study was a Māori-led partnership between academic researchers and tāngata whaikaha Māori communities, grounded in a Kaupapa Māori Research paradigm. We have shown that it is possible to measure these things in a more culturally relevant way for the Māori population and generate robust descriptive data that will form the basis for comprehensive analysis.

The large sample size (7320 participants) enables us to account for heterogeneity within the Māori population and diverse experiences of wellbeing and disability across six culturally relevant dimensions (Te Pae Māhutonga domains) not previously described in a study of this scale. The nature and extent of the data collected necessitate that we present only the high-level findings. More detailed analysis, including an exploration of the intersection between different dimensions of Māori identity and experiences and the relationship with power structures (e.g., racism, ableism, and disablism), will be discussed in future publications.

This study used Māori descent for the sampling frame, rather than self-identified Māori ethnicity for the following reasons. Firstly, descent takes a tāngata whenua, indigenous rights-based approach; secondly, descent data provide a sampling frame more sensitive to the long-term effects of colonization (progressive acculturalization); thirdly, the use of descent aligns with iwi Māori data collection principles; finally, the electoral roll, which we used for recruitment, captures Māori descent, not ethnicity. Our results show that 4.6% of participants of Māori descent do not identify as being of Māori ethnicity. Had we collected ethnicity instead of descent, we would have missed this sizeable group (*n* = 333 in the study and *N* = 16,230 in Māori descent population).

Ethnicity is a social construct of cultural affiliation and is a marker of social perception which influences people’s experiences and outcomes; is pertinent to understanding and measurement of health and equity; and self-identification aligns with principles of self-determination and the right for Māori to name themselves as Māori [28]. Māori, as tāngata whenua, also have international human rights (UNDRIP), which recognize the right of indigenous peoples to self-determination and to be recognized collectively as Indigenous [29]. NZ domestic law confers specific legal rights to Māori relating to ownership of land and natural resources, cultural preservation, and political representation [28]. Most NZ statutes use ancestry criteria to define who is Māori, as the closest concept to whakapapa (genealogy), to determine who can enrol in a Māori electorate, make a claim under The Treaty of Waitangi Act 1975, and own Māori land.

Mauriora: Access to Te Ao Māori

We found that 32.3% of participants are able to understand, at least fairly well, spoken te reo Māori when it is spoken by others. Given that our survey was able to be completed in either te reo Māori or English, this is not likely to be an underestimate. This result aligns with data from Te Kupenga Survey 2018, which showed that 29.4% of respondents could speak it at least fairly well [9]. Protective effects of language on Indigenous wellbeing are well-documented [30]. However, despite its position as a valued possession of Māori [31] and an official language since 1987, most Māori participants report that they are not able to speak, write, or understand Māori very well, or not at all. Although the NZ Native Schools Act 1867 banning the use of te reo in schools was repealed in 1979, widespread use of te reo has yet to be achieved. The government needs to develop policies that would assist in the revival of te reo to maximise health gains from the protective effect of language.

2.Waiora: Environmental Resources

Connections to whakapapa, whenua, marae access, and traditional practices were explored under the Waiora domain as critical resources for the preservation of Māori wellbeing. Autonomy and social connection have been demonstrated to mitigate the negative effects of socioeconomic disadvantage and discrimination and enable fuller participation in society [32]. Our results show that, for 97.3% of participants, whānau Māori is much more than the nuclear family, extending into multiple generations, with 46.7% of participants feeling very connected to their whānau. Over 45% of participants also included kaupapa whānau (friends) as part of their whānau. This aligns with Kukutai [22], who reported that kaupapa whānau made up almost 50% of those whom Māori considered whānau. The collective notion of whakapapa whānau is a cornerstone of Māori society and continues to be a critical component of wellbeing that extends beyond those who live in the same household [22,33].

This sense of connection as whānau has been regarded by Māori as a critical element of the concept of whānau wellbeing and has arguably provided a level of resilience to decades of discriminatory oppression since colonization [34]. Māori view community resilience through the lens of collective responsibility; each whānau holds specific responsibilities contributing to the collective function and survival of the community or iwi [35]. Consequently, given the diversity of whānau contexts, the use of a narrow definition of family for policymaking is problematic and could contribute to further inequities for Māori [36]. Reluctance across government agencies to fully resource whānau ora concepts contributes to perpetuating inequities and fails to capitalize on the inherent strength and opportunities of te ao Māori to build health gains for Māori.

3.Toiora: Healthy Lifestyles

Life satisfaction and health-related quality of life are often used interchangeably in research, policy, and practice; however, these terms are connected to different theoretical concepts and need to be reported separately [37]. Life satisfaction involves conscious appraisal of life and includes emotional reactions to life events, whereas health-related quality of life is the match between a person’s self-rated health, independence, and social connections and their normative sociocultural value system [38]. This conceptual difference is highlighted in our results by discordant responses to these two questions; 49.9% of participants reported feeling satisfied or very satisfied with their life; however, 71.4% reported having at least good, very good, or excellent general health. These findings potentially demonstrate either internalized racism/ableism leading to the normalization of poor health outcomes for Māori, or significant negative contribution of non-health-related issues to overall life satisfaction.

Most health-related QoL measures do not incorporate Indigenous models of wellbeing, including domains prioritized by Indigenous peoples such as culture, spirituality, whānau, and connection to the land, nor elements, such as the experience of colonization, dispossession, and assimilation [39] that have been demonstrated to negatively impact Indigenous populations. The omission of these factors reduces the cross-cultural validity of these data [40] and likely inflates reported quality of life results.

4.Te Oranga: Participation in Society

Te Oranga describes the extent to which people are free from experiencing discrimination, to what extent they feel included in society, and whether they have reliable access to quality healthcare. Over 50% of participants reported discrimination in the last 12 months; many experienced multidimensional discrimination. Factors such as race (74.8%), skin colour (49.9%), and appearance (45.5%) were the most frequently perceived reasons for the discrimination. The primary care patient survey showed that 22.7% of Māori experience discrimination [41]. Our higher discrimination rate may be related to the approach used to collect the data, but further in-depth analysis is required. The most frequent setting for discrimination was school (68.7%), followed by ‘on the street’ (60.4%), at work (55.4%), and within the judicial system (35.2%). Experiences of discrimination within the education system have been reported previously [42], with accounts suggesting that the discrimination is usually by teachers and staff rather than fellow students.

Measures of self-reported discrimination in NZ reveal persistent structural and institutional discrimination, resulting in longstanding inequities in education, employment, income, justice, and health outcomes [43]. Structural discrimination also contributes to poorer social inclusion for marginalized groups. Discrimination fosters distrust in public services and reinforces exclusionary practices, limiting meaningful participation in community life and broader societal opportunities. In NZ, it also represents a breach of the guarantees of Te Tiriti o Waitangi by the government. For Māori, this systemic bias mirrors the challenges faced in healthcare, where policies perpetuate social isolation and diminished access to culturally responsive support systems [44]. Only 20.6% of participants reported feeling included ‘a lot’ within their community, 13.0% within Te Ao Māori, and 16.5% within broader New Zealand. According to Kitching et al., Indigenous populations face significant disparities due to systemic discrimination in healthcare exacerbated by inconsistent access to primary care [45]. The results of this survey reflect this finding; 32% of participants reported having an unmet health need and 32.6% cited the cost of accessing health as the reason.

5.Ngā Manukura: Leadership

This domain explores leadership opportunities such as employment status, occupation, and the level of involvement in environmental activities. Over 10% of participants were involved in iwi or hapū environmental planning and leadership. Due to limited comparative data, it is unclear how this value compares with other population groups; however, these data are consistent with the Household Labour Force Survey 2018 on organization-based volunteering among Māori (13.1%), which is higher than that of all ethnicities at 12.4%. Associations between health and wellbeing and community leadership are well-documented. Community volunteering improves happiness in older wāhine Māori [46]. Conversely, cultural or iwi leadership roles can present additional burdens if such roles bring additional workload pressures, leading to poorer wellbeing outcomes [47]. To counteract this, Māori, as kaitiaki (guardians), should be supported and resourced to fulfil cultural leadership roles which ensure protection of environmental, economic, cultural, and spiritual wellbeing.

Over 50% of participants reported being employed. The benefits of meaningful work have been shown to be effective in multiple ways, especially the expression of reaching one’s potential [48]. Among those employed, the highest proportion of people worked as professionals, followed by technicians and trade workers, labourers, then managers. In contrast, IDI data revealed higher proportions of managers, then professionals, labourers, and service workers [49]. This discrepancy emphasizes the potential inaccuracy of existing government survey and administrative data and the need to supplement it with other data sources that have been designed by and for Māori.

6.Te Mana Whakahaere: Autonomy

This domain examines the degree to which autonomy and self-determination are linked to economic freedom and material wellbeing for Māori. Material hardships were evident amongst the survey responses. Nearly a quarter of participants (24.4%) reported that they had to borrow from family or friends to meet everyday living costs. Participants reported economizing by buying cheaper cuts or less meat (60.1%) and less fresh fruit and vegetables (34.8%), and 36.2% reported having insufficient reserves to be able to incur an unexpected cost of NZD 500.

Financial strain not only limits the choices available to whānau to do the things that bring them satisfaction and lift their wairua (spirits), but such limitations also have downstream effects on social, cultural, economic, and behavioural wellbeing and can create psychosocial stress, unmet need, and intergenerational harm. These impacts are widely recognized as being associated with profound life-course effects, especially among children and young adults [50,51], and are mediated through complex pathways arising from social stratification via differential exposures to risk, vulnerability, and consequences that perpetuate social and life-course inequities [51]. They erode sustainability at an individual, community, and iwi level, resulting in loss of autonomy. This sustainability, along with resilience, is key to our future wellbeing and depends on strong community connections and access to adequate resources [52]. Our findings highlight the need for government policies and programs focusing on material living conditions to improve health and wellbeing outcomes for Māori and increase the ability of whānau and communities to be self-determining.

### Implications for Indigenous People Globally

This research aimed to focus on Māori, as opposed to existing representative, non-indigenous surveys. We have demonstrated that an Indigenous-led survey, incorporating Indigenous methodology and interpretation, provides additional and more culturally relevant insights into health, wellbeing and disability. We argue that this type of approach is not only essential for understanding the experiences of Indigenous subpopulations globally, but it also has the potential to enable more targeted policies that effectively address existing inequities.

## 5. Strengths and Limitations

This is the first large-scale, nationally representative survey of Māori that provides a comprehensive assessment of holistic wellbeing cognizant of Māori and disability diversity. It addresses information gaps about Māori disability in a way that is shaped by Māori disability communities themselves and creates the tools to monitor outcomes beyond the accessibility and content limitations of Crown data sources. We were able to achieve a response rate comparable to that in other surveys using an electoral roll sample frame [53,54]. Given our high survey completion rate and the weighting of results to key variables, our data can be considered representative of the Māori descent population [14].

Previous reporting of limitations related to the survey methodology [14] recognized that the sampling frame, contact process, and response format are all sources of potential selection bias in the study results. The electoral rolls are the most comprehensive population sampling frames readily available to researchers, but they include only those aged over 18 years and enrolled to vote. While our data were weighted first to the electoral roll and then to the APC, differences in the coverage of both these resources are likely to vary by socioeconomic position and sex. It is not possible to quantify the impact of non-sampling bias, but the impact is likely to be the underestimation of results associated with low socioeconomic status, especially for men. Thus, our results showing poor social and health outcomes are likely to be underestimates; those relating to good outcomes are likely to be overestimates.

In the absence of culturally validated survey instruments for Indigenous Māori, our approach was to use a combination of whole-population and Māori-specific official statistical tools [14], thus enabling comparability with published population data, then adding questions sourced from the Māori disability community. Results for Māori in this survey may not be directly comparable with those of other government surveys, despite using largely identical survey items. This is due to the national surveys being administered at different time points, using different sampling methodologies; they also report by Māori ethnicity, rather than Māori descent, resulting in different denominator populations.

## 6. Conclusions

We used a Māori health promotion framework to present the data, as government surveys and other forms of data collection have failed to provide a comprehensive picture of Māori well-being from a Te Ao Māori perspective. As Crown decision-making relies increasingly on integrated data infrastructure, it is imperative that data collected be valid and reliable, or the policy developed through use of the data gathered will be flawed. Our model has highlighted gaps in most of the standardized questionnaire instruments, but we have demonstrated that culturally relevant and accessible data collection is possible through a national survey. Further analyses will examine the intersectionality of Māori experiences and their relationship with systemic power structures. These results can be used to fill critical data gaps for invisible populations, such as tāngata whaikaha Māori. Doing so would fulfil part of the guarantees to Māori under Te Tiriti o Waitangi and New Zealand’s human rights obligations. 

## Figures and Tables

**Table 1 ijerph-22-00829-t001:** Participants—demographics (study population *n* = 7230 and estimated total (APC Māori descent adult) population *N* = 526,426).

Variable	Categories	Estimated Population Proportion			Estimated Population Total		
		Percent (%)	Lower CI	Upper CI	Number of People	Lower CI	Upper CI
Gender ^†‡^	Male (Tāne)	46.70	45.20	48.30	246,000	235,000	257,000
	Female (Wāhine)	50.70	49.10	52.20	267,000	259,000	274000
	Non-binary genders (Total)	1.10	0.70	1.40	5550	3700	7410
	Trans male (Tangata ira tāne)	0.2 ** [N = 4]	-	-	<1000 **	-	-
	Trans female (Whakawāhine)	0.1 ** [N = 9]	-	-	<1000 **	-	-
	Takatāpui	0.5 ** [N = 29]	-	-	2500 **	-	-
	Other gender	0.3 ** [N = 13]	-	-	1500 **	-	-
	No response	1.60	1.20	1.90	8300	6490	10,100
Age-Group ^†‡^	18–29	29.90	28.20	31.50	157,000	146,000	168,000
	30–44	26.10	24.80	27.40	137,000	130,000	145,000
	45–54	14.90	14.00	15.70	78,200	74,100	82,300
	55–64	12.60	11.90	13.30	66,500	63,100	69,900
	65–74	6.90	6.40	7.30	36,200	34,000	38,400
	75+	3.40	3.10	3.80	18,000	16,300	19,800
	No response	6.30	5.50	7.00	33,000	28,800	37,100
Ethnicity *	European	56.4	54.9	57.9	297,000	287,000	307,000
	Māori	95.4	94.8	96.0	502,000	492,000	513,000
	Pacific Peoples	7.0	6.1	7.9	36,700	31,900	41,500
	Asian	1.7	1.3	2.1	9080	6990	11200
	MELAA	-	-	-	-	-	-
	Other	6.1	5.4	6.8	32,200	28,700	35,700
Region ^†‡^	Northland	7.50	6.70	8.20	39,300	35,100	43,500
	Auckland	23.50	22.20	24.80	124,000	117,000	131,000
	Waikato	13.70	12.60	14.80	72,000	65,800	78,100
	Bay of Plenty	11.60	10.50	12.60	60,800	55,300	66,300
	Gisborne	2.90	2.30	3.40	15,000	11,900	18,100
	Hawke’s Bay	5.50	4.80	6.20	29,100	25,300	32,900
	Taranaki	3.20	2.60	3.70	16,800	13,800	19,700
	Manawatū-Whanganui	7.00	6.20	7.80	36,900	32,800	41,100
	Wellington	10.10	9.20	10.90	53,100	48,600	57,600
	Tasman	0.90	0.60	1.20	4600	3140	6060
	Nelson	0.70	0.40	0.90	3560	2360	4760
	Marlborough	0.80	0.50	1.00	4010	2700	5330
	West Coast	0.60	0.30	0.90	3190	1840	4550
	Canterbury	7.80	7.00	8.50	40,900	36,900	44,900
	Otago	2.90	2.50	3.40	15,500	13,100	17,900
	Southland	1.50	1.20	1.80	7770	6210	9340
Urbanicity	Major urban	60.20	58.70	61.70	317,000	307,000	327,000
	Minor urban	23.10	21.80	24.40	121,000	114,000	129,000
	Rural	16.30	15.20	17.30	85,800	80,200	91,400
	No response	0.40	0.20	0.60	2130	1180	3090

Study population n = 7230 and estimated total (APC Māori descent adult) population N = 526,426. * For Ethnicity and Iwi, multiple answers were allowed, so these columns *do not* sum to 100%. ^†^ These variables have been standardized to the electoral roll. ^‡^ These variables have been standardized to the APC. ** These response categories had very small response rates; thus, the estimates have low precision.

**Table 2 ijerph-22-00829-t002:** Participants—Iwi affiliations.

Variable	Categories	Estimated Population Proportion			Estimated Population Total		
		Percent (%)	Lower CI	Upper CI	Number of People	Lower CI	Upper CI
Iwi *	Te Tai Tokerau/Tāmaki-Makaurau (Northland/Auckland) Region Iwi	26.72	25.36	28.08	141,000	133,000	148,000
	Hauraki (Coromandel) Region Iwi	2.19	1.73	2.65	11,500	9090	13,900
	Waikato/Te Rohe Pōtae (Waikato/King Country) Region Iwi	10.38	9.46	11.30	54,600	49,700	59,600
	Te Arawa/Taupō (Rotorua/Taupō) Region Iwi	7.74	6.95	8.53	40,800	36,500	45,000
	Tauranga Moana/Mātaatua (Bay of Plenty) Region Iwi	14.93	13.83	16.03	78,600	72,600	84,600
	Te Tai Rāwhiti (East Coast) Region Iwi	12.00	11.03	12.97	63,200	57,900	68,400
	Te Matau-a-Māui/Wairarapa (Hawke’s Bay/Wairarapa) Region Iwi	7.89	7.06	8.73	41,600	37,100	46,100
	Taranaki Region Iwi	6.55	5.82	7.28	34,500	30,600	38,400
	Whanganui/Rangitīkei (Wanganui/Rangitīkei) Region Iwi	3.27	2.72	3.82	17,200	14,300	20,200
	Manawatū/Horowhenua/Te Whanganui-a-Tara (Manawatū/Horowhenua/Wellington) Region Iwi	4.65	4.08	5.22	24,500	21,500	27,500
	Te Wai Pounamu (South Island) Region Iwi	11.43	10.53	12.33	60,200	55400	65,000
	Rēkohu/Wharekauri (Chatham Islands) Region Imi/Iwi	0.38	0.22	0.54	1990	1160	2830

Study population n = 7230 and estimated (APC Māori descent adult) population N = 526,426. * For Ethnicity and Iwi (tribe), multiple answers were allowed, so these columns *do not* sum to 100%.

**Table 3 ijerph-22-00829-t003:** Mauriora—Te Reo Māori.

Variable	Categories	Estimated Population Proportion			Total Population		
		Percent (%)	Lower CI	Upper CI	Number of People	Lower CI	Upper CI
Te reo Māori—How well are you able to speak Māori in day-to-day conversation?	Cannot speak any Te Reo Māori	8.6	7.7	9.4	45,100 (40,600–49,600)	40,600	49,600
	Only a few words or phrases	44.5	42.9	46.0	234,000	225,000	244,000
	Not very well	29.0	27.6	30.3	153,000	145,000	160,000
	Fairly well	10.3	9.4	11.2	54,400	49,400	59,300
	Well	3.7	3.1	4.3	19,700	16,600	22,800
	Very well	2.9	2.4	3.4	15,200	12,500	17,900
	No response	1.0	0.8	1.3	5460	4000	6920
How well are you able to understand spoken Māori?	No more than a few words or phrases	29.3	27.9	30.7	154,000	146,000	162,000
	Not very well	36.3	34.8	37.7	191,000	183,000	199,000
	Fairly well	19.7	18.5	20.9	104,000	97,100	110,000
	Well	8.6	7.7	9.5	45,300	40,600	50,000
	Very well	4.9	4.2	5.6	25,800	22,200	29,400
	No response	1.3	1.0	1.6	6710	5010	8410
How well are you able to read Māori, with understanding?	No more than a few words or phrases	33.2	31.8	34.7	175,000	166,000	184,000
	Not very well	35.8	34.4	37.3	189,000	180,000	197,000
	Fairly well	16.9	15.8	18.1	89,200	82,900	95,500
	Well	7.4	6.6	8.2	39,100	34,900	43,200
	Very well	5.3	4.5	6.0	27,700	23,900	31,400
	No response	1.3	1.0	1.6	6920	5250	8590
How well are you able to write in Māori, with understanding?	No more than a few words or phrases	42.6	41.1	44.2	225,000	215,000	234,000
	Not very well	35.1	33.7	36.5	185,000	177,000	193,000
	Fairly well	12.6	11.6	13.6	66,200	60,800	71,600
	Well	4.8	4.2	5.5	25,400	21,900	29,000
	Very well	3.4	2.8	3.9	17,600	14,600	20,600
	No response	1.5	1.2	1.8	7830	6120	9550
How easy is it to access Māori language learning resources?	Very difficult	3.6	3.0	4.2	18,800	15,600	21,900
	Difficult	10.7	9.8	11.7	56,500	51,300	61,700
	Neither easy nor difficult	37.7	36.2	39.2	199,000	189,000	208,000
	Easy	29.6	28.3	31.0	156,000	149,000	164,000
	Very easy	16.2	15.1	17.3	85,500	79,500	91,400
	No response	2.1	1.7	2.4	11,000	9200	12,900

**Table 4 ijerph-22-00829-t004:** Mauriora—cultural participation.

Variable	Categories	Estimated Population Proportion			Estimated Population Total		
		Percent (%)	Lower CI	Upper CI	Number of People	Lower CI	Upper CI
Been to a Māori festival or event, such as Pā Wars, Matariki, or Waitangi Day celebrations? (last 12 months)	Yes/No (% positive)	42.7	41.2	44.2	225,000	216,000	234,000
Sung a Māori song, performed a haka, or given a mihi or speech? (last 12 months)	Yes/No (% positive)	61.9	60.4	63.4	326,000	316,000	336,000
Taken part in a kapa haka as a performer, either competitive or social? (last 12 months)	Yes/No (% positive)	15.3	14.2	16.5	80,600	74,200	87,000
Provided any unpaid help or skills for a kapa haka performer, group, or event? (last 12 months)	Yes/No (% positive)	15.1	14.1	16.2	79,600	74,000	85,200
Taken part in other Māori performing arts or crafts? (last 12 months)	Yes/No (% positive)	18.8	17.6	20.0	99,000	92,600	105,000
Said a karakia?	Yes/No (% positive)	57.8	56.3	59.3	304,000	294,000	314,000
Been to a hui? (last 12 months)	Yes/No (% positive)	48.8	47.3	50.3	257,000	248,000	266,000
Taken part in traditional Māori healing or massage? (last 12 months)	Yes/No (% positive)	14.7	13.7	15.7	77,400	71,800	82,900
Been to other activities on a marae? (last 12 months)	Yes/No (% positive)	45.0	43.5	46.5	237,000	228,000	246,000

**Table 5 ijerph-22-00829-t005:** Waiora—connection to environmental and cultural resources.

Variable	Categories	Estimated Population Proportion			Estimated Population Total		
		Percent (%)	Lower CI	Upper CI	Number of People	Lower CI	Upper CI
Do you know your iwi or tribe?	Yes/No (% positive)	89.2	88.2	90.2	470,000	459,000	480,000
Do you know your hapū or sub-tribe?	Yes/No (% positive)	61.4	59.9	62.9	323,000	313,000	333,000
Do you know your maunga or mountain?	Yes/No (% positive)	70.7	69.3	72.1	372,000	362,000	383,000
Do you know your awa, moana, river, or water?	Yes/No (% positive)	69.3	67.8	70.7	365,000	355,000	375,000
Do you know your waka or canoe?	Yes/No (% positive)	64.0	62.5	65.5	337,000	327,000	347,000
Do you know your tīpuna/tūpuna, or ancestor?	Yes/No (% positive)	62.0	60.5	63.5	327,000	317,000	336,000
Do you know your marae tīpuna/tūpuna, or ancestral marae?	Yes/No (% positive)	70.0	68.5	71.4	368,000	358,000	378,000
How connected do you feel to your tūrangawaewae?	I feel not at all connected	22.7	21.4	24.0	120,000	112,000	127,000
	I feel very weakly connected	11.3	10.4	12.3	59,600	54,400	64,800
	I feel weakly connected	12.6	11.5	13.6	66,100	60,200	72,000
	I feel somewhat connected	19.6	18.4	20.8	103,000	96,800	110,000
	I feel strongly connected	14.1	13.1	15.1	74,300	68,800	79,800
	I feel very strongly connected	16.4	15.3	17.5	86,300	80,300	92,400
	No response	3.3	2.8	3.8	17,100	14,500	19,800
Marae Access							
Have you ever been to a marae?	Yes/No (% positive)	96.0	95.5	96.6	506,000	495,000	516,000
Have you been to a marae in the last 12 months?	Yes/No (% positive)	48.0	46.5	49.5	253,000	243,000	262,000
About how many times in the last 12 months have you been to a marae?	Once	10.6	9.6	11.5	55,700	50,500	60,900
	Twice	10.9	9.9	11.8	57,300	52,100	62,400
	3–5 times	13.8	12.8	14.9	72,700	67,000	78,400
	6–10 times	6.3	5.6	7.1	33,400	29,300	37,500
	11–20 times	2.4	2.0	2.8	12,700	10,500	14,900
	More than 20 times	3.9	3.4	4.4	20600	17,900	23,300
	No response	52.1	50.6	53.6	274000	264000	284,000
Have you ever been to any of your ancestral marae?	Yes	59.3	57.8	60.8	312,000	302,000	322,000
About how many times in the last 12 months have you been to your ancestral marae?	Once	9.5	8.6	10.4	49,900	44,900	54,900
	Twice	6.0	5.3	6.7	31,700	28,100	35,200
	3–5 times	7.0	6.2	7.8	36,800	32,600	40,900
	6–10 times	3.3	2.7	3.8	17,100	14,200	20,100
	11–20 times	1.5	1.2	1.8	8030	6370	9690
	More than 20 times	2.5	2.1	3.0	13,300	11,000	15,700
	Not visited in the last 12 months	66.6	65.2	68.1	351,000	340,000	361,000
	No response	3.6	3.0	4.1	18,800	16,000	21,600
Cultural Practices							
In the last 12 months, did you gather any traditional Māori food, such as kaimoana, eel, or pikopiko?	Yes/No (% positive)	37.7	36.2	39.2	198,000	189,000	207,000
In the last 12 months, did you gather any materials for use in traditional Māori practices, such as weaving or rongoā?	Yes/No (% positive)	20.6	19.4	21.8	109,000	102,000	115,000
Gathered either traditional food or materials for traditional practices	Yes/No (% positive)	44.7	43.2	46.2	235,000	226,000	245,000

**Table 6 ijerph-22-00829-t006:** Waiora—Connection to Whānau.

Variable	Categories	Estimated Population Proportion			Estimated Population Total		
		Percent (%)	Lower CI	Upper CI	Number of People	Lower CI	Upper CI
Which group or groups include the people you think of about as your whānau?	My parents, partner/spouse, brothers/sisters, brothers-in-law/sisters-in-law, children	97.3	96.9	97.8	512,000	502,000	523,000
	My grandparents, grandchildren	77.5	76.3	78.8	408,000	397,000	419,000
	My aunts and uncles, cousins, nephews/nieces, other in-laws	80.4	79.2	81.6	423,000	413,000	434,000
	My friends, others	45.9	44.4	47.4	242,000	232,000	251,000
In general, how would you rate how your whānau is doing these days?	Not at all well	1.3	0.9	1.8	7090	4790	9400
	Not very well	8.8	7.9	9.7	46,400	41,500	51,300
	Somewhat well	63.7	62.2	65.1	335,000	325,000	346,000
	Very well	25.0	23.8	26.3	132,000	125,000	139,000
	No response	1.1	0.9	1.4	6020	4510	7530
How connected do you feel to your whānau?	Not at all connected	1.5	1.1	1.9	7710	5610	9800
	Not very connected	9.8	8.8	10.8	51,500	46,100	57,000
	Somewhat connected	41.0	39.5	42.5	216,000	207,000	225,000
	Very connected	46.7	45.2	48.2	246,000	237,000	255,000
	No response	1.0	0.7	1.3	5240	3780	6700
In the last 4 weeks, how often have you seen any whānau (who don’t live with you)?	Not seen in the last 4 weeks	16.8	15.7	18.0	88,500	82,300	94,800
	About once a month	14.0	13.0	15.1	73,900	68,200	79,500
	About once a fortnight	15.6	14.5	16.7	82,200	76,100	88,300
	About 1 to 2 days a week	23.3	22.0	24.6	123,000	115,000	130,000
	About 3 to 4 days a week	12.5	11.5	13.5	65,800	60,500	71,100
	Every day or almost every day	17.0	15.8	18.1	89,300	82,800	95,900
	No response	0.8	0.6	1.0	4220	2970	5480
Would you say that you have too much contact, about the right amount of contact, or not enough contact with whānau (who don’t live with you)?	Not enough	44.2	42.7	45.7	233,000	223,000	242,000
	About the right amount	53.2	51.6	54.7	280,000	270,000	290,000
	Too much	1.8	1.3	2.3	9440	6730	12,200
	No response	0.9	0.6	1.1	4610	3310	5910

**Table 7 ijerph-22-00829-t007:** Toiora—life satisfaction, general health, disability, and activity limitation.

Variable	Categories	Estimated Population Proportion			Estimated Population Total		
		Percent (%)	Lower CI	Upper CI	Number of People	Lower CI	Upper CI
Life Satisfaction							
How do you feel about your life right now?	Very dissatisfied	3.3	2.6	3.9	17,300	13,900	20,700
	Dissatisfied	12.3	11.2	13.4	64,600	58,500	70,700
	Neither satisfied nor dissatisfied	27.3	25.9	28.7	144,000	136,000	152,000
	Satisfied	35.9	34.4	37.3	189,000	181,000	197,000
	Very satisfied	14.0	13.1	14.9	73,600	68,800	78,400
	No response	7.3	6.5	8.1	38,400	33,900	42,900
General Health							
In general, would you say your health is?	Poor	5.0	4.3	5.6	26,200	22,600	29,800
	Fair	18.3	17.1	19.5	96,100	89,400	103,000
	Good	33.9	32.5	35.4	179,000	170,000	187000
	Very good	27.8	26.4	29.1	146,000	139,000	153,000
	Excellent	9.7	8.8	10.6	51,100	46,200	56,000
	No response	5.4	4.6	6.1	28,300	24,300	32,200
Self-Reported Disability							
Do you have a disability, long-term condition, or mental health condition that limits your ability to carry out everyday activities?	Yes/No (% positive)	21.2	20	22.4	112,000	105,000	118,000
Activity Limitation							
Washington Group Short Set on Functioning (WG-SS)	‘A lot’ of limitation or ‘Cannot do at all’ to any of: seeing, hearing, walking, remembering/concentrating, self-care, or communicating	15.1	14	16.2	79,400	73,300	85,500
	Either self-reported disability OR activity limitation	28.1	26.7	29.5	148,000	140,000	156,000

**Table 8 ijerph-22-00829-t008:** Toiora—health-related quality of life.

Variable	Categories	Estimated Population Proportion			Estimated Population Total		
		Percent (%)	Lower CI	Upper CI	Number of People	Lower CI	Upper CI
Physical Health							
During the past 4 weeks, how much of the time have you accomplished less than you would like, as a result of your physical health?	All of the time	4.7	4.0	5.3	24,700	21,200	28,200
	Most of the time	13.7	12.6	14.7	71,900	66,200	77,600
	Some of the time	24.5	23.2	25.8	129,000	121,000	136,000
	A little of the time	24.7	23.4	26.0	130,000	123,000	137,000
	None of the time	26.8	25.4	28.1	141,000	134,000	148,000
	No response	5.7	5.0	6.5	30,200	26,100	34,300
During the past 4 weeks, how much of the time have you been limited in the kind of work or other regular daily activities you do, as a result of your physical health?	All of the time	4.1	3.5	4.7	21,600	18,300	24,900
	Most of the time	9.5	8.6	10.4	50,000	45,300	54,800
	Some of the time	20.0	18.8	21.2	105,000	98,500	112,000
	A little of the time	23.3	22.0	24.5	122,000	116,000	129,000
	None of the time	37.4	35.9	38.9	197,000	188,000	206,000
	No response	5.7	5.0	6.5	30,100	26,000	34,200
Mental Health							
During the past 4 weeks, how much of the time have you accomplished less than you would like, as a result of your emotional health, such as feeling depressed or anxious?	All of the time	4.1	3.4	4.8	21,600	17,800	25,400
	Most of the time	10.2	9.2	11.2	53,700	48,300	59,000
	Some of the time	21.0	19.8	22.3	111,000	104,000	118,000
	A little of the time	23.8	22.5	25.1	125,000	118,000	133,000
	None of the time	35.0	33.6	36.4	184,000	177,000	192,000
	No response	5.9	5.1	6.6	30,800	26,700	35,000
During the past 4 weeks, how much of the time have you done work or other regular daily activities less carefully than usual, as a result of your emotional health?	All of the time	2.4	1.9	2.9	12,600	9870	15,400
	Most of the time	6.8	6.0	7.6	35,900	31,600	40,200
	Some of the time	19.2	17.9	20.4	101,000	93,800	108,000
	A little of the time	25.0	23.6	26.3	131,000	124,000	139,000
	None of the time	40.7	39.2	42.2	214,000	206,000	222,000
	No response	6.0	5.2	6.7	31,400	27,200	35,600
During the past 4 weeks, how much has pain interfered with your normal work, including both work outside the home and housework?	Extremely	2.8	2.4	3.3	15,000	12,400	17,600
	Quite a bit	9.7	8.8	10.6	51,000	46,100	55,900
	Moderately	13.8	12.7	14.9	72,500	66,500	78,600
	A little bit	31.6	30.2	33.1	167,000	159,000	175,000
	Not at all	36.4	34.9	37.8	192,000	183,000	200,000
	No response	5.6	4.9	6.4	29,700	25,600	33,700
How much of the time during the past 4 weeks have you felt calm and peaceful?	None of the time	3.9	3.2	4.6	20,500	17,000	24,100
	A little of the time	18.8	17.5	20.0	98,900	91,800	106,000
	Some of the time	30.0	28.6	31.4	158,000	150,000	166,000
	Most of the time	36.8	35.3	38.2	193,000	185,000	202,000
	All of the time	4.9	4.3	5.6	26,000	22,600	29,400
	No response	5.6	4.9	6.4	29,700	25,700	33,700
How much of the time during the past 4 weeks have you had a lot of energy?	None of the time	6.6	5.9	7.4	34,800	30,800	38,800
	A little of the time	22.8	21.5	24.1	120,000	113,000	127,000
	Some of the time	37.0	35.5	38.5	195,000	186,000	203,000
	Most of the time	25.1	23.8	26.4	132,000	125,000	139,000
	All of the time	2.8	2.3	3.3	14,800	11,900	17,600
	No response	5.7	4.9	6.4	29,800	25,800	33,800
How much of the time during the past 4 weeks have you felt downhearted or depressed?	All of the time	1.8	1.4	2.3	9640	7270	12,000
	Most of the time	9.8	8.8	10.8	51,700	46,000	57,300
	Some of the time	19.9	18.7	21.2	105,000	98,000	112,000
	A little of the time	32.9	31.5	34.3	173,000	165,000	181,000
	None of the time	29.7	28.4	31.1	157,000	149,000	164,000
	No response	5.8	5.0	6.5	30,300	26,300	34,400
How much of the time during the past 4 weeks have your physical or emotional health interfered with your social activities, like visiting friends or relatives?	All of the time	5.1	4.3	5.8	26,600	22,400	30,800
	Most of the time	11.7	10.7	12.8	61,700	56,000	67,400
	Some of the time	18.9	17.7	20.1	99,400	92,900	106,000
	A little of the time	22.9	21.6	24.2	121,000	113,000	128,000
	None of the time	35.5	34.1	37.0	187,000	179,000	195,000
	No response	5.9	5.1	6.7	31,100	26,900	35,300

**Table 9 ijerph-22-00829-t009:** Te Oranga—Healthcare Access.

Variable	Categories	Estimated Population Proportion			Estimated Population Total		
		Percent (%)	Lower CI	Upper CI	Number of People	Lower CI	Upper CI
Access to GP Care							
Do you have a GP clinic or medical centre that you usually go to when you are feeling unwell or are injured?	Yes/No (% positive)	88.0	86.9	89.1	463,000	453,000	474,000
Have you contacted your usual medical centre, for your own health, in the past 12 months?	Yes/No (% positive)	74.0	72.5	75.4	390,000	380,000	399,000
In the past 12 months, has there been a time when you wanted to speak to a GP, nurse, or other health care worker at your usual medical centre, within the next 24 h, but they were unable to speak to you?	Yes/No (% positive)	34.6	33.2	36.1	182,000	174,000	399,000
In the past 12 months, has there been a time when you had a medical problem, but did not contact a GP because of cost?	Yes/No (% positive)	32.6	31.2	34.1	172,000	163,000	191,000
Access to Dental Care							
How long has it been since you last visited a dental health care worker about your own dental health, for any reason?	I have never visited a dental health care worker	3.1	2.6	3.6	16,200	13,600	18,900
	5 or more years ago	24.0	22.7	25.3	126,000	119,000	134,000
	More than 2 years ago but less than 5	18.7	17.5	19.9	98,300	91,500	105,000
	More than 1 year ago but less than 2	16.8	15.7	18.0	88,600	82,100	95,000
	Within the past year	30.9	29.6	32.3	163,000	155,000	170,000
	No response	6.5	5.7	7.3	34,300	30,000	38,600
In the past 12 months, have you avoided going to a dental health care worker because of cost?	Yes/No (% positive)	59.1	57.6	60.5	311,000	300,000	322,000
In the past 12 months, did you ever feel that you needed professional help for your emotions, stress, mental health, or substance use, but you did not receive that help?	Yes/No (% positive)	22.0	20.6	23.4	116,000	108,000	124,000

**Table 10 ijerph-22-00829-t010:** Te Oranga—social inclusion.

Variable	Categories	Estimated Population Proportion			Estimated Population Total		
		Percent (%)	Lower CI	Upper CI	Number of People	Lower CI	Upper CI
How included do you feel within your community?	Not at all	21.8	20.5	23.1	115,000	107,000	122,000
	A little	47.3	45.8	48.8	249,000	240,000	258,000
	A lot	20.6	19.5	21.8	109,000	103,000	115,000
	No response	10.3	9.3	11.3	54,100	48,700	59,500
How included do you feel within Te Ao Māori?	Not at all	30.2	28.8	31.7	159,000	151,000	167,000
	A little	42.6	41.1	44.1	224,000	216,000	233,000
	A lot	13.0	12.1	14.0	68,500	63,500	73,500
	No response	14.1	13.0	15.3	74,400	68,200	80,600
How included do you feel within broader New Zealand?	Not at all	20.9	19.6	22.2	110,000	103,000	118,000
	A little	48.0	46.5	49.5	253,000	243,000	262,000
	A lot	16.5	15.5	17.5	87,000	81,600	92,400
	No response	14.6	13.4	15.7	76,600	70,500	82,800
How much impact does the way New Zealand is structured (its policies, services, supports, attitudes) have on your ability to do day-to-day activities?	None at all	16.5	15.5	17.6	87,100	81,300	92,900
	A little	40.7	39.2	42.2	214,000	205,000	223,000
	A lot	29.6	28.3	31.0	156,000	148,000	164,000
	No response	13.1	12.0	14.2	69,100	62,900	75,300
How much impact does the way New Zealand is structured (its policies, services, supports, attitudes) have on your ability to achieve your life goals?	None at all	17.2	16.1	18.3	90,400	84,400	96,500
	A little	36.9	35.4	38.3	194,000	186,000	202,000
	A lot	33.5	32.1	35.0	177,000	168,000	185,000
	No response	12.4	11.3	13.5	65,300	59,400	71,200

**Table 11 ijerph-22-00829-t011:** Te Oranga—discrimination.

Variable	Categories	Estimated Population Proportion			Estimated Population Total		
		Percent (%)	Lower CI	Upper CI	Number of People	Lower CI	Upper CI
Have you ever been discriminated against?	Yes/No (% positive)	58.4	56.9	59.9	308,000	297,000	318,000
What do you think the discrimination was based on?	Skin colour?	49.9	47.9	52.0	154,000	145,000	162,000
	Race or ethnic group?	74.8	73.0	76.6	230,000	221,000	240,000
	Gender?	29.3	27.5	31.0	90,000	84,100	95,900
	Age?	23.6	21.9	25.3	72,500	66,800	78,200
	A disability or health issue you have?	13.5	12.1	14.9	41,400	36,900	46,000
	Sexual orientation?	7.6	6.4	8.9	23,500	19,500	27,500
	Religious beliefs?	11.5	10.1	12.8	35,300	30,800	39,700
	Your income or your whānau’s income?	25.7	23.8	27.5	79,000	72,600	85,300
	Your appearance?	45.5	43.5	47.6	140,000	132,000	149,000
	Something else?	9.4	8.2	10.6	28,800	25,000	32,600
Where did the discrimination occur?	At school?	68.7	67.0	70.5	211,000	202,000	221,000
	Trying to get a job?	39.9	37.9	41.9	123,000	116,000	130,000
	At work?	55.4	53.3	57.4	170,000	162,000	179,000
	Trying to get housing or a mortgage?	29.0	27.2	30.8	89,200	82,800	95,600
	Dealing with the police or the courts?	35.2	33.3	37.2	108,000	101,000	116,000
	Trying to get medical care?	31.4	29.5	33.3	96,600	89,900	103,000
	Trying to get service in a shop or restaurant?	44.4	42.3	46.4	137,000	129,000	144,000
	On the street or in a public place?	60.4	58.5	62.3	186,000	177,000	195,000
	In any other situation?	52.1	50.1	54.1	160,000	152,000	169,000
Did any of the discrimination happen in the last 12 months?	Yes/No (% positive)	53.6	51.6	55.6	165,000	156,000	174,000

**Table 12 ijerph-22-00829-t012:** Ngā Manukura—community leadership.

Variable	Categories	Estimated Population Proportion			Estimated Population Total		
		Percent (%)	Lower CI	Upper CI	Number of People	Lower CI	Upper CI
Community Leadership							
In the last 12 months, did you take part in any environmental planning on behalf of your iwi, hapū, or marae?	Yes/No (% positive)	11.2	10.3	12.2	59,100	54,000	64,100
Professional Leadership							
In employment?	Yes/No (% positive)	52.0	50.4	53.5	274,000	265,000	282,000
Occupation ^†^							
	Managers	6.7	6.1	7.2	35,000	32,200	37,900
	Professionals	10.5	9.8	11.1	55,100	52,100	58,100
	Technicians and Trades Workers	8.4	7.4	9.4	44,300	38,800	49,800
	Community and Personal Service Workers	5.8	5.2	6.4	30,500	27,400	33,700
	Clerical and Administrative Workers	6.3	5.7	6.8	32,900	30,100	35,700
	Sales Workers	2.5	2.1	2.9	13,100	11,000	15,300
	Machinery Operators and Drivers		3.3	4.5	20,700	17,300	24,000
	Laborers	8.0	7.1	8.8	41,900	37,100	46,700
	Students	11.5	10.5	12.6	60,600	54,800	66,400
	Retirees	3.5	3.2	3.8	18,400	17,000	19,900
	Others Not In Labor Force	13.2	12.1	14.3	69,400	63,200	75,600
	Not Stated	19.0	17.7	20.4	100,000	92,400	108,000
	No response	0.8	0.5	1.1	4320	2790	5850

^†^ These variables have been standardized to the electoral roll.

**Table 13 ijerph-22-00829-t013:** Te Mana Whakahaere—socioeconomic opportunities.

Variable	Categories	Survey Population			Estimated Population Total		
		Percent (%)	Lower CI	Upper CI	Number of People	Lower CI	Upper CI
NZDep2018 ^†,‡^							
	Lowest deprivation	9.9	9.2	10.7	52,300	48,700	56,000
	Second quintile	12.3	11.5	13.2	65,000	60,500	69,400
	Third quintile	16.4	15.3	17.5	86,200	80,400	92,000
	Fourth quintile	22.2	21.0	23.4	117,000	110,000	124,000
	Highest deprivation	38.7	37.2	40.3	204,000	194,000	214,000
	No response	0.4	0.2	0.6	2130	1180	3090
Individual Income							
	Loss	0.5	0.3	0.7	2640	1540	3750
	Zero income	2.7	2.2	3.2	14,300	11,500	17,000
	NZD 1–5000	4.3	3.6	5.1	22,800	18,800	26,900
	NZD 5001–10,000	3.6	3.0	4.3	19,100	15,700	22,500
	NZD 10,001–15,000	5.3	4.6	6.1	28,000	23,900	32,100
	NZD 15,001–20,000	6.4	5.7	7.2	33,800	29,800	37,800
	NZD 20,001–25,000	5.9	5.2	6.7	31,100	27,200	35,100
	NZD 25,001–30,000	4.7	4.1	5.3	24,800	21,600	27,900
	NZD 30,001–35,000	3.9	3.3	4.4	20,400	17,500	23,400
	NZD 35,001–40,000	4.6	4.0	5.3	24,400	21,000	27,800
	NZD 40,001–50,000	8.1	7.2	8.9	42,400	37,800	47,000
	NZD 50,001–60,000	8.7	7.8	9.6	45,700	41,100	50,400
	NZD 60,001–70,000	7.8	7.0	8.6	41,000	36,700	45,400
	NZD 70,001–100,000	11.5	10.6	12.4	60,600	55,800	65,500
	NZD 100,001–150,000	6.0	5.4	6.6	31,600	28,500	34,700
	NZD 150,001 or more	2.8	2.5	3.2	15,000	13,100	16,800
	No response	13.0	12.0	14.1	68,600	63,000	74,200

^†^ These variables have been standardized to the electoral roll. ^‡^ These variables have been standardized to the APC.

**Table 14 ijerph-22-00829-t014:** Te Mana Whakahaere—material wellbeing (income adequacy).

Variable	Categories	Estimated Population Proportion			Estimated Population Total		
		Percent (%)	Lower CI	Upper CI	Number of People	Lower CI	Upper CI
Ownership or participation							
Do you have a good bed?	Yes/No (% positive)	81.7	80.4	82.9	430,000	420,000	440,000
Do you have a meal with meat, fish, or chicken (or vegetarian equivalent), at least every second day?	Yes/No (% positive)	86.4	85.3	87.5	45,5000	444,000	465,000
Do you have two pairs of shoes in a good condition that are suitable for your daily activities?	Yes/No (% positive)	83.8	82.5	85.0	441,000	431,000	451,000
Do you have suitable clothes for important or special occasions?	Yes/No (% positive)	79.8	78.5	81.1	420,000	410,000	430,000
Do you have home contents insurance?	Yes/No (% positive)	53.3	51.8	54.9	281,000	272,000	289,000
Do you have access to a car or van for personal use?	Yes/No (% positive)	83.0	81.7	84.2	437,000	42,7000	447,000
Do you usually replace worn-out clothes with new clothes, rather than second-hand?	Yes/No (% positive)	61.7	60.2	63.2	325,000	315,000	335,000
Do you have access to both a computer and an internet connection at home?	Yes/No (% positive)	83.6	82.4	84.8	440,000	430,000	451,000
Do you have a get-together with friends or extended family for a drink or meal at least once a month?	Yes/No (% positive)	61.7	60.2	63.2	325,000	315,000	335,000
Do you give presents to family or friends on birthdays, Christmas, or other special occasions?	Yes/No (% positive)	78.1	76.7	79.4	411,000	401,000	421,000
Do you usually have a holiday away from home for at least a week every year?	Yes/No (% positive)	47.2	45.7	48.7	248,000	240,000	257,000
Do you have a holiday overseas at least every three years?	Yes/No (% positive)	24.2	23.0	25.4	127,000	121,000	134,000

**Table 15 ijerph-22-00829-t015:** Te Mana Whakahaere—material wellbeing (economizing, financial freedoms, and financial capacity).

Variable	Categories	Estimated Population Proportion			Estimated Population Total		
		Percent (%)	Lower CI	Upper CI	Number of People	Lower CI	Upper CI
Economising							
In the last 12 months, have you gone without fresh fruit and vegetables, to keep costs down?	Yes/No (% positive)	34.8	33.3	36.3	183,000	174,000	192,000
In the last 12 months, have you bought cheaper cuts of meat or bought less meat than you would have liked, to keep costs down?	Yes/No (% positive)	60.1	58.6	61.6	316,000	306,000	327,000
In the last 12 months, have you postponed or put off visits to the doctor, to keep costs down?	Yes/No (% positive)	38.2	36.7	39.7	201,000	192,000	211,000
In the last 12 months, have you postponed or put off visits to the dentist, to keep costs down?	Yes/No (% positive)	56.6	55.1	58.1	298,000	288,000	308,000
In the last 12 months, have you done without or cut back on trips to the shops or other local places, to keep costs down?	Yes/No (% positive)	57.9	56.5	59.4	305,000	295,000	316,000
In the last 12 months, have you spent less on hobbies or other special interests than you would have liked, to keep costs down?	Yes/No (% positive)	64.3	62.9	65.7	339,000	328,000	349,000
In the last 12 months, have you put up with feeling cold, to keep costs down?	Yes/No (% positive)	36.3	34.7	37.8	191,000	182,000	200,000
In the last 12 months, have you delayed replacing or repairing broken or damaged appliances, to keep costs down?	Yes/No (% positive)	44.3	42.8	45.8	233,000	223,000	243,000
In the last 12 months, have you delayed replacing or repairing broken or worn-out furniture, to keep costs down?	Yes/No (% positive)	44.3	42.8	45.8	233,000	224,000	243,000
Freedoms/Restrictions							
If you (or your partner) had an unexpected and unavoidable expense of NZD 500 in the next week, could you pay that amount within a month without borrowing?	Yes/No (% positive)	63.8	62.3	65.3	336,000	326,000	345,000
Financial Strain							
In the last 12 months, have you been unable to pay electricity, gas, rates, or water bills on time?	Yes/No (% positive)	19.0	17.7	20.3	100,000	92,900	107,000
In the last 12 months, have you been unable to pay for car insurance, registration, or warrant of fitness on time?	Yes/No (% positive)	24.5	23.1	25.9	129,000	121,000	137,000
In the last 12 months, have you been unable to pay the rent or mortgage on time?	Yes/No (% positive)	12.1	11.0	13.2	63,500	57,400	69,600
In the last 12 months, have you been unable to pay hire purchase or other loan payments on time?	Yes/No (% positive)	16.1	14.9	17.4	85,000	77,900	92,000
In the last 12 months, have you borrowed from family or friends to meet everyday living costs?	Yes/No (% positive)	24.4	22.9	25.8	128,000	120,000	137,000
In the last 12 months have you received help in the form of food, clothes, or money from a welfare/community organization such as a church or food bank?	Yes/No (% positive)	18.3	17.0	19.5	96,100	89,100	103,000

## Data Availability

The original contributions presented in this study are included in the article. Further inquiries can be directed to the corresponding author.

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
