# Peer review of "Māori Health, Wellbeing, and Disability in Aotearoa New Zealand: A National Survey"

_ijerph, 2025, doi:10.3390/ijerph22060829_

Round 1
Reviewer 1 Report
Comments and Suggestions for Authors
This interesting study can be considered for publication in IJERPH after the following revisions are made by the authors:
The abstract should start with a background statement, and only then, the study’s aims should be indicated. The obtained results and their implications from a worldwide perspective should be addressed.
Lines 48-50: References are missing.
Line 113: The study’s aims should make part of the Introduction and not be included in a separate section/subsection.
More citations should be included in the Introduction to better support the relevance and the need of the current work.
The number of study participants should be indicated in the Results and not in Materials and Methods.
Lines 157-172: This part of the Methodologies has to be clarified.
Subsections need to be numbered.
I believe that some information in the Material and Methods section could be synthesized and better explained in a clear way.
15 tables are too much. The authors need to decrease this number to 6/7. Some of the information could be given as supplementary material or the current tables can be merged.
The Discussion could be improved by taking into consideration the worldwide relevance that this study can have, and a Conclusions section should be provided separately.
Author Response
COMMENT 1: The abstract should start with a background statement, and only then, the study’s aims should be indicated. The obtained results and their implications from a worldwide perspective should be addressed.
RESPONSE: Background statement and global implications statements added per reviewer’s request.
COMMENT 2: Lines 48-50: References are missing.
RESPONSE: Reference added
COMMENT 3: Line 113: The study’s aims should make part of the Introduction and not be included in a separate section/subsection.
RESPONSE: added to introduction
COMMENT 4: More citations should be included in the Introduction to better support the relevance and the need of the current work.
RESPONSE: we have added an additional 7 citations to the introduction to better support the rationale as suggested.
COMMENT 5: The number of study participants should be indicated in the Results and not in Materials and Methods.
RESPONSE: Number of participants ARE already presented in Results (Lines 329-341), as requested by the reviewer. Data on our methodological sampling framework and sample size calculations are presented in Materials and Methods (e.g. Study Deign - Line 137-142; Study Size - Lines 323-327) in accordance with STROBE Statement Guidelines, and per our previous methodology publication. We have added URL reference to the methodology manuscript.
COMMENT 6: Lines 157-172: This part of the Methodologies has to be clarified.
RESPONSE: These lines of the manuscript are descriptors of two elements of the Ngā Pae Māhutonga model of Māori Health (Durie et al, 1999): Ngā Manukura - Leadership, and Te Mana Whakahaere - Autonomy. They are presented to contextualise the scope of variable selection and domain attribution process required by the STROBE Checklist regarding study variables: "Clearly define all outcomes, exposures, predictors, potential confounders, and effect modifiers."
COMMENT 7: Subsections need to be numbered.
RESPONSE: We thank the Reviewer for this suggestion to provide further clarity and consistency. We have numbered each domain of the model (1 to 6) and cross-refenced those by numbering each subsection in materials and methods, Results, and Discussion with the corresponding Domain reference.
COMMENT 8: I believe that some information in the Material and Methods section could be synthesized and better explained in a clear way.
RESPONSE: We are unclear which elements of the Material and Methods section the reviewer would like our attention. We have once again reviewed this section as a team for clarity and brevity. We had no substantive amendments to our original text to offer.
COMMENT 9: 15 tables are too much. The authors need to decrease this number to 6/7. Some of the information could be given as supplementary material or the current tables can be merged.
RESPONSE: The fundamental premise of this paper is to present the diverse array of related and intersectional issues underpinning the holistic assessment of indigenous well-being, and to present the prevalence of the determinants thereof. Therefore, we consider the full breadth of tabular information to be essential to the body of the results and prefer not to appendicise these critical tables.
We initially structured the tables as one-per-domain of the Pae Mahutonga model (i.e. six tables), but this did not allow for page-fitting to A4, and therefore we reformatted for page layout considerations. We would be very willing to reaggregate to 1 table per domain if the journal accepts the pagination issue of some tables running over to a second page (i.e. reducing the total number of tables to eight).
COMMENT 10: The Discussion could be improved by taking into consideration the worldwide relevance that this study can have, and a Conclusions section should be provided separately.
RESPONSE: Discussion - While we have taken steps to book-end this manuscript in a global indigenous context, we had explicitly limited our discussion to the multifaceted issues affecting Māori. This was for two main reasons, 1. the prioritorisation of Māori as the indigenous peoples of Aotearora (New Zealand), with mana (inherent prestige) in our own right. We sought to avoid historic colonial impositions of Māori as merely a comparator population. 2. We recognise this manuscript is already lengthy. For space constraints we elected to prioritise the comprehensive exposition of Māori context and relating our findings to other (domestic) literature affecting Māori specifically.
However, we acknowledge that these results will be of great interest to other Indigenous populations globally, therefore, we have taken the reviewers suggestion and have added additional discussion (Lines 621-629).
RESPONSE: Conclusion Section – Added
Reviewer 2 Report
Comments and Suggestions for Authors
Overview
This article is the analysis of data drawn from a national Indigenous centred survey. This study uses Te Pae Mahutonga, a Māori health promotion framework to position their analysis within a Te Ao Māori perspectives on health, well-being and disability. These findings provide nuanced detail of Māori perspectives of health, wellbeing and disability. This study is innovative and novel advancing Indigenous centred and aligned approaches to survey methodology. An important study which demonstrates that such survey approaches are feasible.
Minor changes
Line 40 – Remove , so the line reads ‘a culturally aligned methodological approach’
Line 53 – Is citation 1 relevant
Line 82 – To reinforce your argument, I would recommend highlighting the weakness of this survey
Page 13 – Table ‘Traditional practice’ – Would you consider using Cultural Practices vs Traditional Practices for the title of this section?
Table 10 Te Oranga - Social Inclusion line 414 – You make no direct commentary on the bespoke social inclusion questions you added. How did you think your newly developed inclusion questions went? Line 549 you state, ‘the level in which they can actively participate.’ This does not reflect your question which asks about their feelings.
Line 467 - Line 42 – As you do not analyse intersectionality and systemic power in this article, do you need this line as you refer to this line 467.
Accept with minor amendments.
Author Response
COMMENT 1: Line 40 – Remove , so the line reads ‘a culturally aligned methodological approach’
RESPONSE: Agree - Done.
COMMENT 2: Line 53 – Is citation 1 relevant
RESPONSE: No. Agree - it had been included in error.
COMMENT 3: Line 82 – To reinforce your argument, I would recommend highlighting the weakness of this survey
RESPONSE: This survey refers to our methodology paper. We have discussed the limitations of the survey methodology therein, however we assert that the present survey is the first large scale, nationwide survey led by Māori researchers that integrates health, wellbeing, and disability data within a Kaupapa Māori framework. We do, if we interpret the reviewers intent correctly, discuss the limitations of the national Government led surveys in the Introduction (Lines 75-94) and Discussion (Lines 651-658)
COMMENT 4: Page 13 – Table ‘Traditional practice’ – Would you consider using Cultural Practices vs Traditional Practices for the title of this section?
RESPONSE: Change Accepted.
COMMENT 5: Table 10 Te Oranga - Social Inclusion line 414 – You make no direct commentary on the bespoke social inclusion questions you added. How did you think your newly developed inclusion questions went? Line 549 you state, ‘the level in which they can actively participate.’ This does not reflect your question which asks about their feelings.
RESPONSE: Social Inclusion Questions - Thank you for noting those new questions. They (as part of the broad suite of domains/questions considered for this survey) underwent extensive qualitative investigation in their development along with whakawhitikōrero (reflexive feedback processes) as presented in an earlier published methodology work. We have not attempted to 'validate' the results of these here, we fully intend to publish this analysis separately - in a multivariate analysis with comparison to other metrics of inclusion, discrimination, and overall well-being - to characterise and contextualise this question set. We have therefore intentionally avoided that type of analysis here.
RESPONSE: Actively Participate - Thank you. We have amended to correctly reflect your accurate observation.
COMMENT 6: Line 467 - Line 42 – As you do not analyse intersectionality and systemic power in this article, do you need this line as you refer to this line 467.
RESPONSE: We are unclear as to the Reviewer’s exact meaning here. We interpret the comment in reference to whether Line 42 duplicates Line 467. [noting line references have shifted in new version - we refer to originals here]. Both sections, do loosely pertain to elements of self-determination, whether in identity (Line 467) - as it relates to ethnicity and indigeneity - on in rights (Line 42) - customary and NZ legislation. However, the key issue we are addressing in Line 42 is that the dominant culture (in this case the Government, as the designer of national statistics and surveys) has historically had the power and privilege to determine the framework and outcomes under which non-dominant and indigenous culture is presented, and control of the narrative around the articulation of those outcomes. We contend (in Line 42) that this represents an issue of construct (in)validity which renders the existing national epidemiological data on Māori health & well-being inferior. We believe this point stands alone from the issues at 467 as a clear rationale for this manuscript, and therefore should be retained.
Reviewer 3 Report
Comments and Suggestions for Authors
Thank you for this important work. The scale and non-government nature are impressive and critical for self-determination and sovereignty.
I am grateful for several parts of this work, including it being Maori-led and also the inclusive gender categories for participants to select from.
I have no comments or suggestions other than a desire for this group to repeat this work across several other aspects of life for Maori.
Author Response
Comment 1: I have no comments or suggestions other than a desire for this group to repeat this work across several other aspects of life for Maori.
RESPONSE: We are truely grateful, and as a team we do have bold plans to do just that.